# Clearance of beta-amyloid is facilitated by apolipoprotein E and circulating high-density lipoproteins in bioengineered human vessels

Jerome Robert[1,2]*, Emily B Button[1,2], Brian Yuen[1,2], Megan Gilmour[1,2], Kevin Kang[1,2], Arvin Bahrabadi[1,2], Sophie Stukas[1,2], Wenchen Zhao[1,2], Iva Kulic[1,2], Cheryl L Wellington[1,2]*

[1]Department of Pathology and Laboratory Medicine, University of British Columbia, Vancouver, Canada; [2]Djavad Mowafaghian Centre for Brain Health, University of British Columbia, Vancouver, Canada

**Abstract** Amyloid plaques, consisting of deposited beta-amyloid (Aβ), are a neuropathological hallmark of Alzheimer's Disease (AD). Cerebral vessels play a major role in AD, as Aβ is cleared from the brain by pathways involving the cerebrovasculature, most AD patients have cerebrovascular amyloid (cerebral amyloid angiopathy (CAA), and cardiovascular risk factors increase dementia risk. Here we present a notable advance in vascular tissue engineering by generating the first functional 3-dimensioinal model of CAA in bioengineered human vessels. We show that lipoproteins including brain (apoE) and circulating (high-density lipoprotein, HDL) synergize to facilitate Aβ transport across bioengineered human cerebral vessels. These lipoproteins facilitate Aβ42 transport more efficiently than Aβ40, consistent with Aβ40 being the primary species that accumulates in CAA. Moreover, apoE4 is less effective than apoE2 in promoting Aβ transport, also consistent with the well-established role of apoE4 in Aβ deposition in AD.
DOI: https://doi.org/10.7554/eLife.29595.001

*For correspondence:
jrme.robert@gmail.com (JR);
wcheryl@mail.ubc.ca (CLW)

**Competing interests:** The authors declare that no competing interests exist.

## Introduction

Alzheimer's Disease (AD) is the leading cause of senile dementia with over 44 million affected persons and an economic burden of over $600 billion (*Mayeux and Stern, 2012*). In addition to AD's neuropathological hallmarks of amyloid plaques consisting of deposited Aβ peptides and neurofibrillary tangles consisting of hyperphosphorylated tau proteins, 60–90% of AD brains have evidence of cerebral amyloid angiopathy (CAA), cerebral small vessel disease and microvascular degeneration (*Attems and Jellinger, 2014*). Apolipoprotein (apo) E, which in the brain is secreted primarily from astrocytes, is the principal lipid carrier within the central nervous system (CNS) and, in humans, exists as three isoforms, namely *APOE2, APOE3,* and *APOE4* (*Lane-Donovan and Herz, 2017*). ApoE has well-established effects on Aβ metabolism with apoE4 being detrimental, apoE3 neutral and apoE2 protective. ApoE is also hypothesized to contribute to cerebrovascular dysfunction (*Zlokovic, 2013*). As the major routes by which Aβ is cleared from the brain involve the cerebrovasculature (*Ueno et al., 2014*), understanding the vascular contributions to dementia is of great interest (*Snyder, 2015*).

Cardiovascular risk factors including type two diabetes mellitus (T2DM), hypertension, hypercholesterolemia, obesity and stroke increase AD risk (*Duron and Hanon, 2008*), yet the mechanisms by which cardiovascular health impacts brain function remain poorly understood. Interestingly,

**eLife digest** Alzheimer's disease causes gradual loss of memory and difficulties in learning. The brains of patients with the disease show several abnormalities including deposits of a peptide molecule called beta-amyloid that is known to be toxic to nerve cells. This peptide can also cause damage to the brain by accumulating within the muscular walls of large blood vessels, a condition known as cerebral amyloid angiopathy (CAA) and is present in most Alzheimer's disease patients.

A group of molecules known as lipoproteins, which transport fats throughout body fluids, are thought to be involved in the process by which beta-amyloid leaves the brain. Apolipoprotein E (apoE) is one such molecule and it is made in the brain by cells called astrocytes. There are three different versions of apoE that are associated with different levels of risk of developing Alzheimer's disease. Other lipoproteins, such as high-density lipoprotein, which is present in the blood, may also play a role in clearing beta-amyloid proteins from the brain. However, it has been difficult to investigate the roles of these lipoproteins in Alzheimer's disease because current test-tube models do not fully mimic the composition of human brain blood vessels or show how they work.

Robert et al. have used a tissue engineering approach to generate the first three-dimensional model of human brain blood vessels that can reproduce cerebral amyloid angiopathy. To make the model, different types of human cells similar to those found in real blood vessels and astrocytes were grown under conditions that resemble real-life conditions, including mimicking blood flow through the engineered vessels. Having established that the engineered vessels behaved similarly to normal blood vessels, Robert et al. used them to test whether lipoproteins helped to clear beta-amyloid proteins from the vessels. These experiments showed that a form of apoE that protects against Alzheimer's disease was more effective in transporting beta-amyloid proteins across the walls of blood vessels than other forms of apoE. Further experiments showed that high-density lipoprotein in the blood and apoE on the brain side of the vessel work together to help transport beta-amyloid into the vessels.

Together, these findings show that the model of CAA developed by Robert et al. provides a valuable new tool for exploring how this condition develops. The model could also be used more widely in the future, for example, to study how to deliver new drugs that could help treat Alzheimer's disease into the brain.

DOI: https://doi.org/10.7554/eLife.29595.002

epidemiological studies suggest that AD risk may be attenuated by high levels of circulating high-density lipoprotein cholesterol (HDL-C), which is also highly associated with reduced cardiovascular disease (CVD) risk (*Zuliani et al., 2010*). Specifically, levels of apoA-I, the major HDL-associated protein, positively correlate with Mini-Mental State Examination (MMSE) and Cognitive Ability Screening Instrument (CASI) scores (*Merched et al., 2000*) and high serum HDL-C levels (>55 mg/dl) in cognitively normal elderly individuals is associated with significantly reduced risk (HR 0.4) of AD even after adjusting for *APOE* genotype and vascular risk factors including obesity and T2DM (*Reitz et al., 2010*). In symptomatic AD patients, plasma apoA-I levels negatively correlate with hippocampal and whole brain volume as well as mean entorhinal cortical thickness (*Hye et al., 2014*), and decreased levels of serum apoA-I can discriminate AD from non-demented age-matched control subjects (*Shih et al., 2014*). As HDL and apoE have several potent vasoprotective functions including reducing inflammation, increasing vascular tone through promoting endothelial nitric oxide (NO) synthase activity, and suppressing vascular adhesion molecule expression (*Stukas et al., 2014b*; *Sacre et al., 2003*), an important goal is to understand how plasma-derived circulating (i.e. HDL) and brain-derived (i.e. apoE) lipoproteins might affect Aβ metabolism in cerebral vessels. However, lipoprotein metabolism in mice and humans are substantially different, as the major circulating lipoprotein in mice is HDL whereas low-density lipoprotein (LDL), which increases CVD risk, is the major circulating lipoprotein in humans (*Yin et al., 2012*). AD animal model studies may therefore not always take into account the inherent vascular resilience of mice compared to humans, and thus may have limits to their translational relevance. Additionally, static in vitro models of human ECs cultured with or without astrocytes do not replicate the complex cellular interactions and extracellular matrix of a native vessel.

To address some of the key limitations of existing experimental models to investigate the role of lipoproteins on Aβ metabolism at the vessel, we generated three-dimensional (3D) bioengineered human vessels using a scaffold-directed dynamic pulsatile flow bioreactor system, where primary human ECs and smooth muscle cells (SMC) were cultivated in the absence or presence of human astrocytes to generate bipartite or tripartite vessels, respectively, which display the histological features of native peripheral and cerebral arteries. Here we demonstrate the utility of this novel experimental platform to investigate how human lipoproteins affect Aβ transport through and accumulation within physiologically relevant bioengineered human vessels.

## Results

### Production and anatomical characterization of bioengineered bipartite vessels

Bipartite bioengineered vessels were fabricated by sequentially seeding primary human myofibroblasts and ECs isolated from umbilical cords into a tubular woven scaffold consisting of polyglycolic acid (PGA), polycaprolactone (PCL) and polylactate (PLA), measuring 15 mm long and 2 mm in diameter (*Figure 1*). After 4 weeks in culture, Haematoxylin-Eosin staining demonstrated the formation of a dense and homogenous tissue on the luminal side of the scaffold (*Figure 2a*). The tissue was composed of cells and extracellular matrix, as demonstrated by the presence of collagen by Picrosirius staining (*Figure 2b*), and confirmed by immunohistochemical detection of collagen IV and laminin in the extracellular matrix (*Figure 2c–d*). Immunohistochemical staining also demonstrated multiple layers of α-smooth muscle actin (α-SMA) positive cells on the inner side of the scaffold and a monolayer of CD31 positive ECs lining the bioengineered vascular lumen (*Figure 2e–f*). Integrity of the endothelial barrier was functionally assessed by injecting Evans blue dye into the bioreactor circulation loop. As expected, Evans blue penetrated into tissue prepared without EC, whereas it was excluded two weeks after EC seeding, demonstrating a functionally tight endothelial barrier (*Figure 2g*). In addition to cord cells, bioengineered vessels could also be fabricated with brain derived SMC (hBSMC) and microvascular cortical EC (hBMEC). Immunohistological staining confirmed that brain-derived and cord-derived primary human cells form similar structures in bioengineered vessels (*Figure 2—figure supplement 1a*), and function similarly with respect to tissue Aβ accumulation and transport (*Figure 2—figure supplement 1b–e*).

### Monomeric aβ accumulates and aggregates in bioengineered bipartite vessels

Accumulation of Aβ within cerebral vessel walls, known as CAA, is a common pathological feature in AD (*Attems and Jellinger, 2014*). To determine if CAA can form in our bioengineered vascular model, we injected monomeric Aβ40 or Aβ42 on the anteluminal side of the vessel to mimic native conditions where Aβ is predominantly produced by neurons. Both ELISA (*Figure 3a,b*) and 6E10 immunostaining (*Figure 3c–f*) confirmed dose-dependent retention of Aβ40 and Aβ42 within the bioengineered vascular wall 48 hr after injection (white), which could be distinguished from autofluorescence by residual scaffold material (shaded in blue). As Aβ fibrillization within the vessel wall is an important feature of CAA, we also stained vessels using Thioflavin-S (Thio-S), and confirmed dose-dependent fibrillization in the bioengineered vessels (*Figure 3e–f*). We further characterized the time course of Aβ accumulation in bioengineered vessels after anteluminal injection of 1 μM monomeric Aβ40 or Aβ42. Both ELISA quantification and 6E10 immunostaining revealed Aβ deposition by 2 hr after injection, after which Aβ levels remained stable for up to 72 hr (*Figure 3g–h*). Interestingly, quantification of beta-sheet formation in bioengineered tissue lysates with Thioflavin-T also revealed increasing signal over time (*Figure 3i–j*), biochemically confirming increased Aβ fibrillization into beta-sheet structures within bioengineered vessels after seeding of Aβ monomers. Finally, extracellular deposition of amyloid was confirmed by immunofluorescent staining for collagen IV, α-SMA, and Aβ. Specifically, confocal microscopy of immunofluorescent staining against amyloid fibrils using the OC antibody demonstrated that fibrillary Aβ accumulates outside of the cells (*Figure 3—figure supplement 1a–b*), whereas 6E10 staining revealed both extracellular accumulation and intracellular vesicular deposition (*Figure 3—figure supplement 1b–f*). Accumulation of Aβ was quantified in tissues engineered from brain- or cord- derived cells, and although deposition of Aβ40 and Aβ42

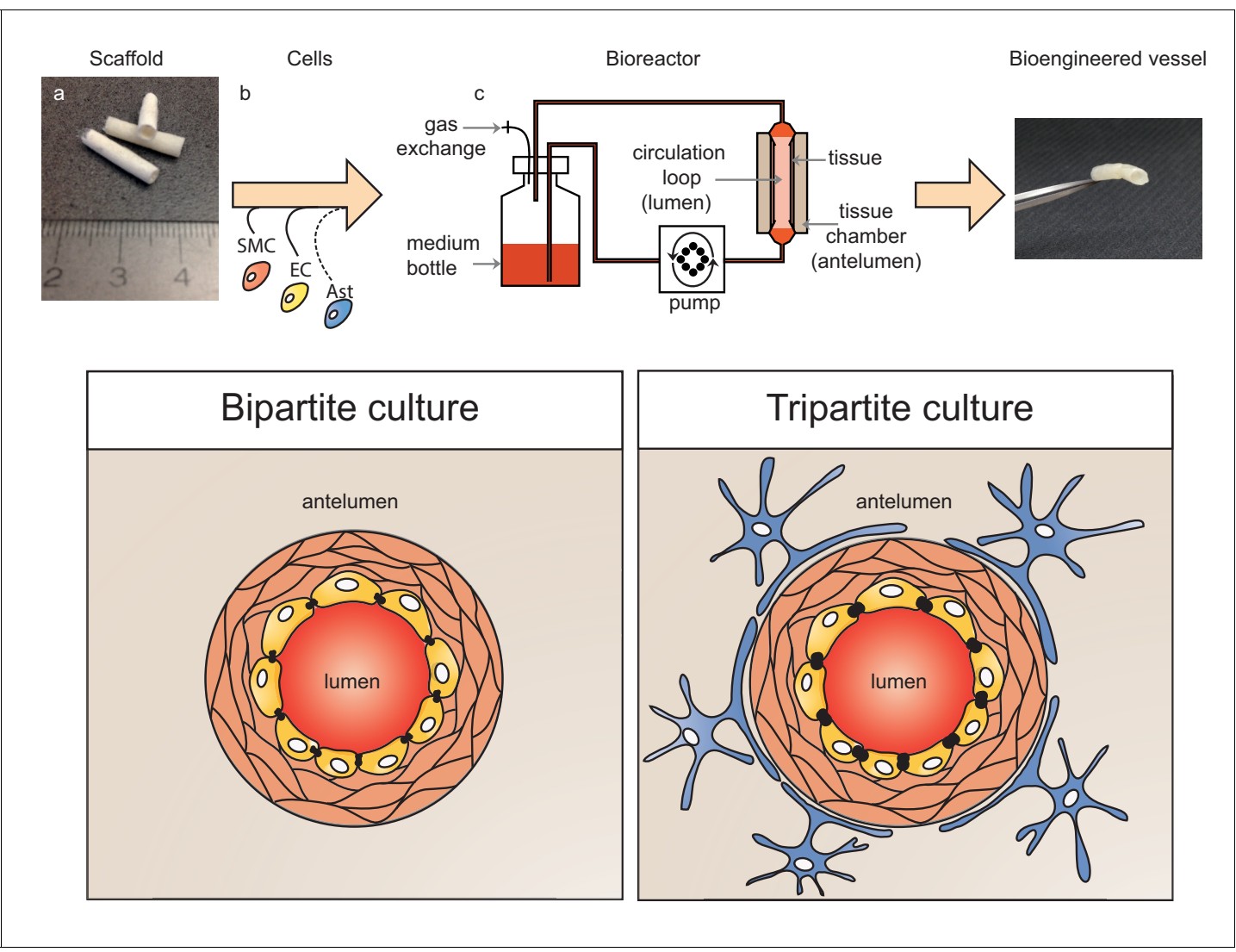

**Figure 1.** Schematic representation of bioengineered vessels. Scaffolding material was prepared into a tubular shape approximately 2 mm in diameter and 15 mm long (**a**). Scaffolds were sequentially seeded with primary human umbilical vein smooth muscle cells (SMC, orange) and endothelial cells (EC, yellow) to form bipartite vessels, or with the addition of primary human astrocytes (Ast, blue) to form tripartite vessels (**b**). Bioengineered vessels were cultivated for approximately 4–6 weeks in a bioreactor containing a tissue chamber (**c**) on the anteluminal side to provide extravascular media, and a circulation loop containing endothelial media that flowed through the vessel lumen under pulsatile conditions using a peristaltic pump.
DOI: https://doi.org/10.7554/eLife.29595.003

tended to be higher in tissues engineered from brain cells, the differences were not significant (*Figure 2—figure supplement 1b–c*). These data provide compelling support that 3D bioengineered human vessels can be used as an in vitro model of CAA.

## Aβ is transported across bioengineered vessels

As a major route of Aβ egress from the brain is direct transport across the cerebral vessel into the circulation (*Ueno et al., 2014*), we next evaluated the suitability of our bioengineered vessels to analyze 'brain-to-blood' Aβ transport by injecting Aβ in the anteluminal tissue chamber compartment of bipartite vessels or scaffold-only controls and measuring the level of Aβ recovered in the circulating medium over 4 hr. ELISA quantification revealed that both Aβ40 and Aβ42 were transported at a slower rate in bioengineered vessels, whereas Aβ freely diffused across scaffold-only controls (*Figure 3k–l*). These data demonstrate the feasibility of bioengineered human vessels to study Aβ recovery in the circulation. Transport of Aβ was also analysed in tissues engineered from brain- or

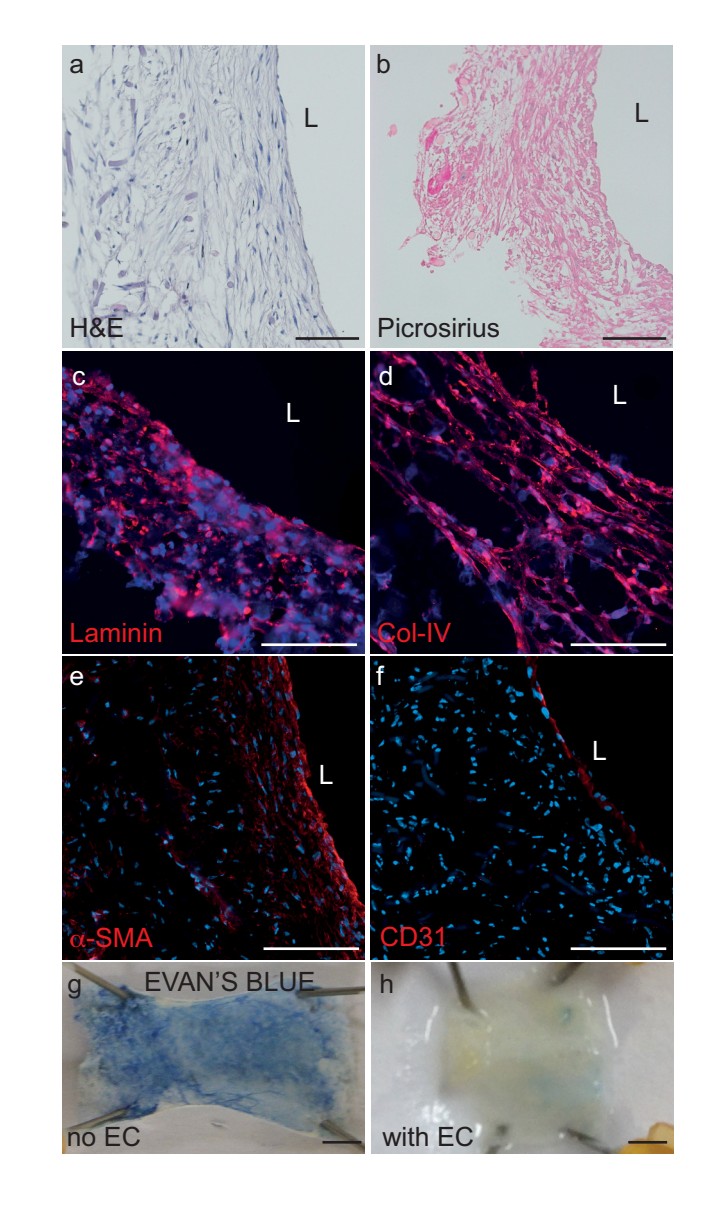

**Figure 2.** Histological structure of bipartite bioengineered vessels. (a) Haematoxylin and Eosin staining revealed dense tissue formation composed of cells and extracellular matrix (ECM). (b) Picrosirius staining confirmed secretion of collagen. Further immunostaining against laminin (c) and collagen IV (d) confirmed ECM secretion by the cells. The expression of α-smooth muscle actin (α-SMA) (e) confirmed the smooth muscle phenotype of the cells in the inner layers, and CD31 positive staining (f) confirmed the presence of an endothelial cell monolayer on the luminal side of the bioengineered vessel. A functional endothelial barrier was confirmed using an Evans blue extravasation assay (g,h). Scale bars represent 200 μm (a–f) or 1 mm (g, h), L: lumen.

DOI: https://doi.org/10.7554/eLife.29595.004

The following figure supplement is available for figure 2:

**Figure supplement 1.** Aβ40 and Aβ42 accumulate similarly in tissues seeded with cells originating from umbilical cord or cortex.

DOI: https://doi.org/10.7554/eLife.29595.005

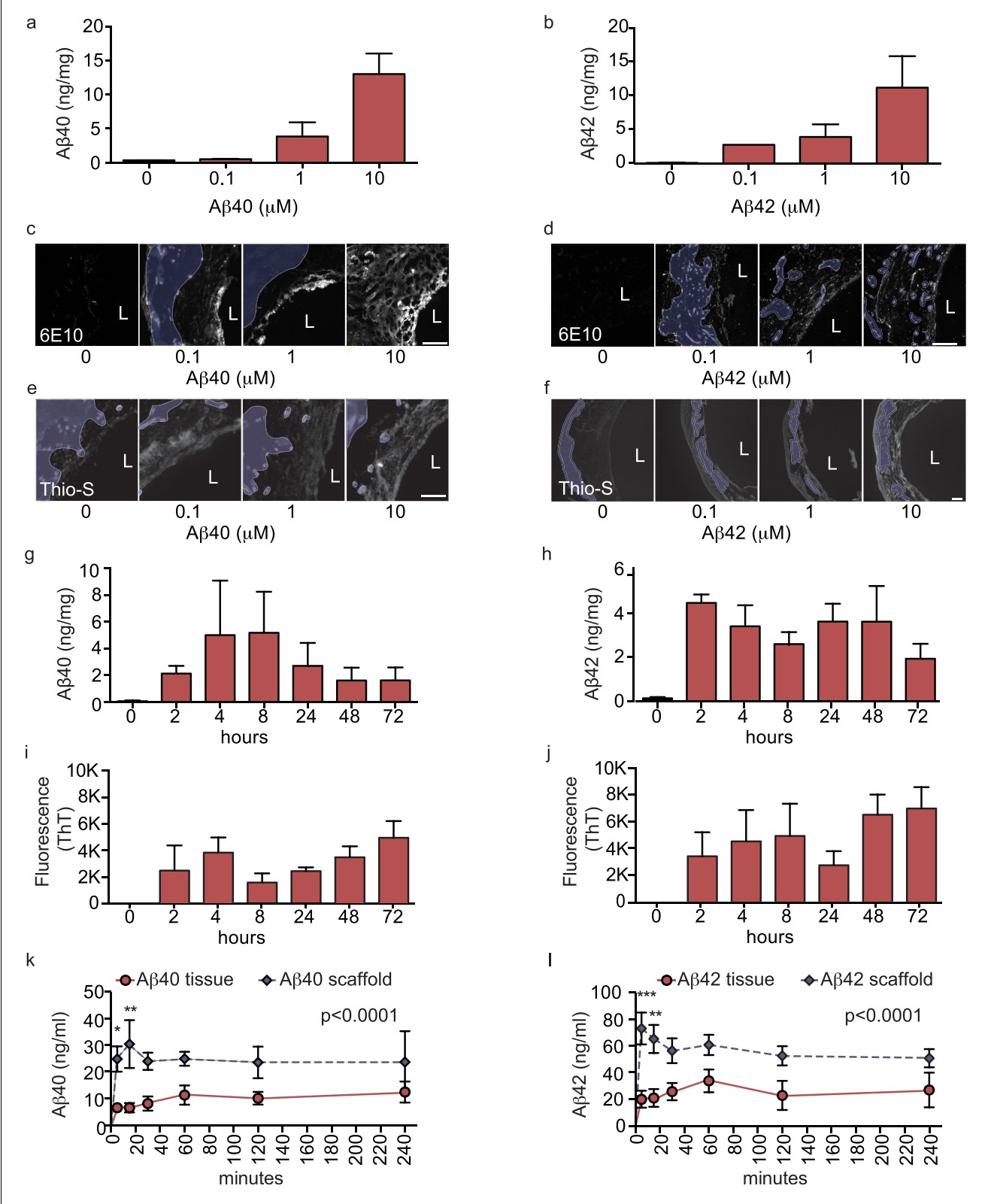

**Figure 3.** Aβ40 and Aβ42 accumulation within and transport through bipartite vessels. Aβ40 and Aβ42 monomers (0, 0.1, 1.0 and 10 µM) were injected into the tissue chamber (antelumen) and incubated for 48 hr under flow conditions (a–f). Aβ deposition within bioengineered vessels was measured using ELISA (a–b), immunostaining with the anti-Aβ antibody 6E10 Aβ (d–e), and Thioflavin-S staining (e,f). To determine the kinetics of CAA formation, Aβ40 and Aβ42 monomers (1 µM) were injected into the tissue chamber and incubated for the indicated times before measuring Aβ tissue

*Figure 3 continued on next page*

*Figure 3 continued*

concentrations by ELISA (**g,h**) or aggregation within the tissue using Thioflavin-T (**i,j**). Aβ transport was measured after injecting Aβ40 and Aβ42 monomers (1 µM) into the anteluminal chamber and sampling media from the circulation (luminal) chamber at the indicated times (**k,l**). Graphs represent mean ±SEM for at least four independent tissues **p=0.01 and ***p=0.001. Bars represent 50 µm, L: lumen.
DOI: https://doi.org/10.7554/eLife.29595.006
The following figure supplement is available for figure 3:

**Figure supplement 1.** Histological structure of Aβ deposited in bioengineered vessels.
DOI: https://doi.org/10.7554/eLife.29595.007

cord- derived cells and demonstrated no significant differences (*Figure 2—figure supplement 1d–e*). Due to the greater ease of obtaining the quantities of primary human cells required to generate over 300 independent tissues used in this study, the remaining experiments were performed with vessels engineered from cord-derived EC and SMC, as they are structurally and functionally equivalent to vessels engineered with brain-derived cells.

## ApoE isoforms differentially reduce amyloid accumulation in bipartite vessels

Genetic variation in *APOE* represents the most common genetic risk factor for AD, with *APOE4* being detrimental and *APOE2* protective (*Zlokovic, 2013*). To test whether apoE regulates vascular Aβ accumulation or transport into the circulation in our engineered bipartite vessels, we injected Aβ40 or Aβ42 monomers in the absence or presence of recombinant apoE of different isoforms into the tissue chamber at a molar ratio 25:1 to mimic the relative concentrations in brain cerebrospinal fluid (CSF) (*Deane et al., 2008*). Although neither apoE isoform significantly altered the rate of recovery of Aβ into the circulation medium over 4 hr (*Figure 4a–b*), we observed a significant and selective decrease of the amount of Aβ42 deposited into the bioengineered tissue after 24 hr with apoE2 co-injection (*Figure 4c–d*).

## Circulating HDL reduces aβ accumulation in bipartite bioengineered vessels and facilitates aβ transport in the presence of anteluminal apoE

As several epidemiological studies associate circulating HDL levels with reduced AD risk (*Zuliani et al., 2010*), and we recently demonstrated that a single intravenous injection of reconstituted HDL (rHDL) acutely lowers soluble brain Aβ levels in APP/PS1 mice (*Robert et al., 2016*), we reasoned that circulating HDL might promote Aβ recovery into the circulation and reduce its accumulation in bioengineered vessels. To test this hypothesis, 200 µg/ml of HDL isolated from normolipidemic young donors were perfused through bioengineered vessels immediately after injecting Aβ into the anteluminal space. Over 4 hr, the levels of Aβ42 and Aβ42 recovered in the circulation medium were slightly increased in the presence of HDL, but this did not reach significance (*Figure 4e,f*). After 24 hr, we observed a strong trend toward decreased tissue levels of accumulated Aβ40 and significantly lower accumulated Aβ42 in the presence of HDL (*Figure 4g–h*).

We then tested for a functional interaction between apoE and HDL by analyzing Aβ transport and tissue accumulation after injecting either beneficial recombinant apoE2 or detrimental recombinant apoE4 into the anteluminal tissue chamber, in the presence or absence of HDL injected into the circulating medium. Importantly, the combination of anteluminal apoE2 and circulating luminal HDL led to significantly increased Aβ40 and Aβ42 transport over 4 hr compared to either Aβ alone or Aβ with apoE2 or HDL alone (*Figure 5a–b*). Consistent with our previous observations at 24 hr, the levels of Aβ40 accumulated in the tissue were not significantly affected by apoE2, HDL, or both apoE and HDL (*Figure 5c*), however, the levels of accumulated Aβ42 were significantly reduced by apoE2, HDL, and the combination of both apoE2 and HDL (*Figure 5d*). Interestingly, the combination of anteluminal apoE4 and circulating luminal HDL significantly increased both Aβ40 and Aβ42 transport over 4 hr compared to either Aβ alone or Aβ with apoE4 (*Figure 5e–f*), and concomitantly the level of Aβ42 accumulated in the tissue at 24 hr was significantly lower in the presence of both apoE4 and HDL compared to Aβ42 alone or Aβ42 and apoE4 (*Figure 5h*). These results strongly support a cooperative role between brain apoE and circulating HDL to preferentially clear Aβ across the

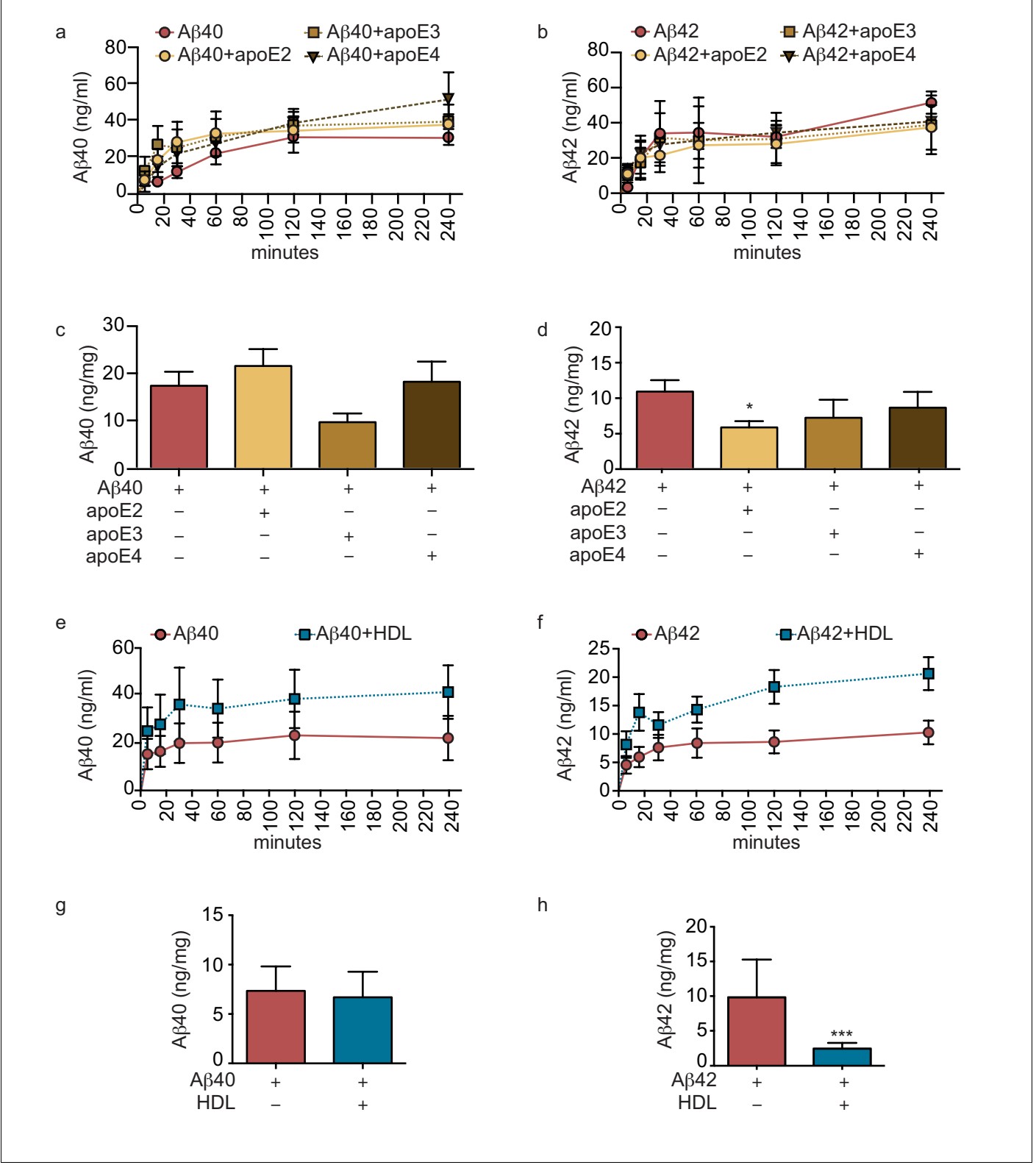

**Figure 4.** Lipoproteins reduce Aβ42 accumulation within bioengineered bipartite vessels. Aβ40 and Aβ42 monomers (1 μM) were incubated without or with recombinant apoE (ratio 25:1) for 2 hr at 37°C before injection into the anteluminal tissue chamber. The levels of transported Aβ was measured by ELISA from samples collected from the luminal circulating medium at the indicated times over 4 hr (a–b), and from vascular tissue collected 24 hr after Aβ injection (c–d). Aβ40 and Aβ42 monomers (1 μM) were injected into the anteluminal chamber in the absence or presence of 200 μg/ml of luminal

*Figure 4 continued on next page*

*Figure 4 continued*

circulating HDL. The levels of transported Ab (**e–f**) and from vascular tissues (**g–h**) were measured as above. Graphs represent mean ±SEM for at least four independent tissues. *p=0.05, **p=0.01 and ***p=0.001.

DOI: https://doi.org/10.7554/eLife.29595.008

vasculature, and suggest that one beneficial role of circulating HDL is to functionally counteract against apoE4 in this Aβ transport assay.

## Structural and functional characterisation of tripartite bioengineered cerebral vessels

Because bipartite vessels consisting of ECs and SMCs have the anatomy of peripheral rather than cerebral vessels, we extended the translational relevance of our bioengineered vessels by incorporating human primary astrocytes on the antelumen (*Figure 1b*) to mimic cerebral vessels. As the diameter of our vessels is approximately 2 mm prior to cell seeding and further contains SMC rather than pericytes, our bioengineered tripartite vessels were specifically designed to resemble larger human leptomeningeal and penetrating arteries rather than the cerebral microvasculature. Tripartite bioengineered vessels are therefore composed of EC lining the lumen, several layers of SMC, and a layer of astrocytes in the antelumen as confirmed by immunofluorescent staining against CD31, α-SMC actin, and GFAP, respectively (*Figure 6a–c*). Interestingly, higher magnification images revealed GFAP positive protrusions penetrating into the tissue (*Figure 6d*), suggestive of astrocyte endfeet. Immunofluorescent staining confirmed the expression of aquaporin four and NDRG2 by astrocytes in bioengineered vessels (*Figure 6—figure supplement 1a–b*). Importantly, we observed that peripherally-derived ECs (HUVEC), when cultured in the presence of astrocytes and under flow conditions, expressed high level of the tight-junction proteins ZO-1 and ZO-2 (*Figure 6e–f*) as well as the blood brain barrier (BBB) tight junction protein Claudin 5 (*Figure 6g*) and the specific BBB transporter Glut-1 (*Figure 6h*), demonstrating that the EC phenotype of a HUVEC is reprogrammed to become brain-like in the presence of astrocytes. Comparison of Glut-1 expression between bipartite, tripartite, umbilical cord and brain tissues confirmed reprograming of HUVEC in engineered vessels, with expression levels comparable to native brain vessels, whereas native umbilical cords lack detectable Glut-1 (*Figure 6—figure supplement 2*). The integrity of the endothelial barrier in tripartite vessels was functionally assessed by injecting Evans blue into the circulation loop and compared to bipartite vessels. As expected, Evans blue was excluded in both tripartite and bipartite vessels (*Figure 6j*). Barrier integrity was further assessed by measuring permeability of 4 kDa or 40 kDa FITC-dextran. Tripartite tissues demonstrated less extravasation of both 4 kDa and 40 kDa than bipartite tissues, although these were not significantly different from each other, whereas, as expected, both tissues are significantly less permeable than unseeded scaffold (*Figure 6—figure supplement 3*). The functionality of EC and astrocytes within tripartite bioengineered vessels was evaluated by measuring nitric oxide (NO) produced by EC and native apoE secretion by astrocytes, which were genotyped as *APOE3/E3*. To determine NO production, vessels were incubated with either 10 nM of acetylcholine (Ach) or 200 µg/ml of HDL for 60 min before measuring conversion of L-$^3$H-arginine to L-$^3$H-citruline. A significant increase was observed after both treatments, whereas, in the presence of the specific NO synthase inhibitor L-NG-nitroarginine methyl ester (L-NAME), the conversion was blocked (*Figure 6k*). For apoE secretion, the brain-penetrant Liver-X-Receptor (LXR) agonist GW3965 (1 µM) was circulated through the lumen of the vessels 72 hr before collecting medium. ELISA quantification demonstrated that GW3965 significantly stimulated the secretion of native apoE in the tripartite vessels (*Figure 6l*). These data confirmed that our tripartite bioengineered vascular tissue has both structural and functional characteristics of native cerebral arteries. Of note, the limited availability of primary human astrocytes with distinct *APOE* genotypes restricted our subsequent experiments in tripartite vessels to *APOE3/E3*.

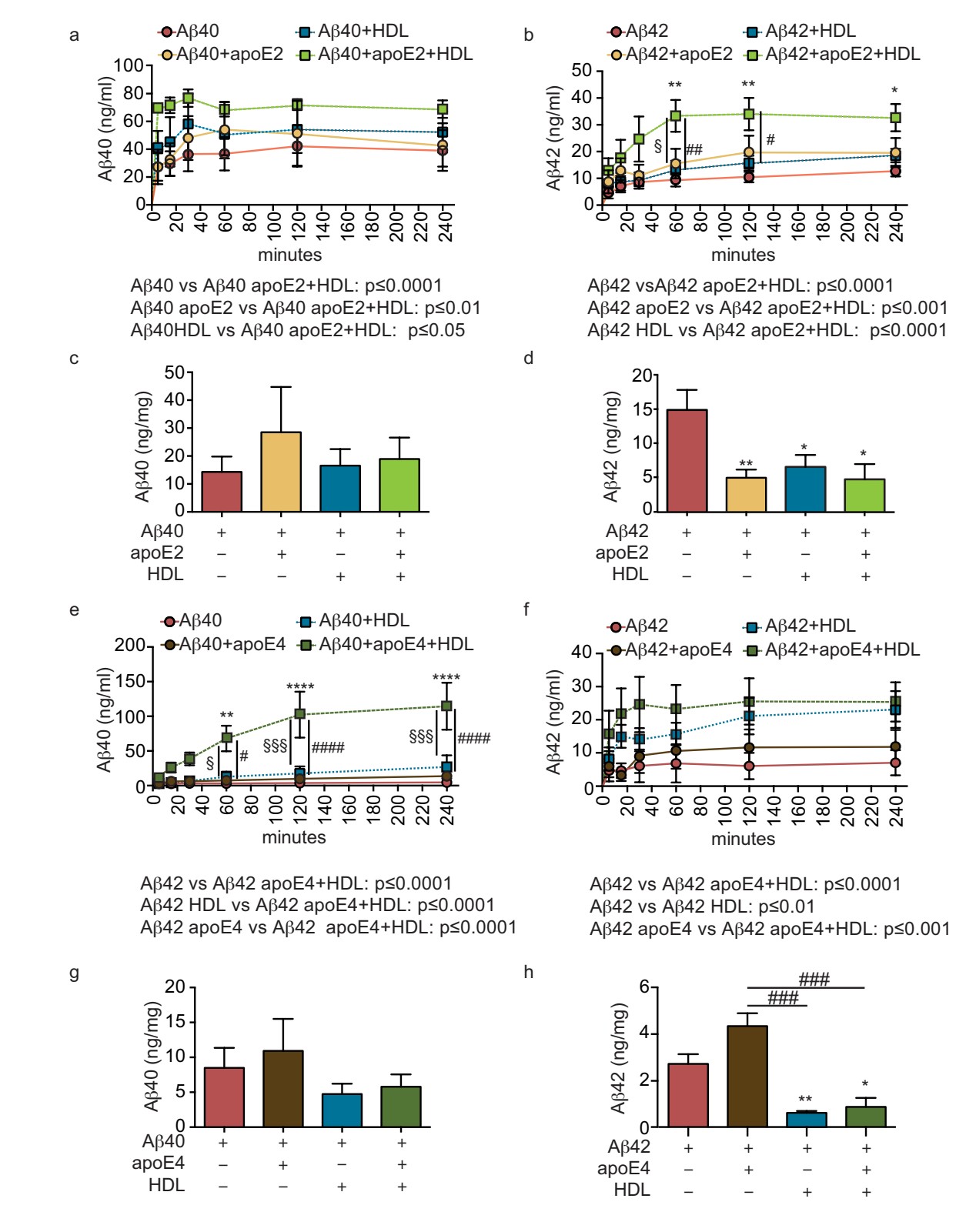

**Figure 5.** HDL facilitates Aβ42 transport and reduces accumulation in bioengineered bipartite vessels in the presence of recombinant apoE. Aβ40 and Aβ42 monomers (1 μM) were incubated without or with recombinant apoE2 (**a–d**) or apoE4 (**e–h**) (ratio 25:1) for 2 hr at 37°C before injecting into the tissue chamber in the absence or presence of 200 μg/ml of circulating HDL, and evaluating transported (**a–b, e–f**)) and accumulated (**c–d, g–h**) by ELISA. Data for Aβ and Aβ +HDL transport presented in *Figure 5a–b,e–f* represent data generated from specific apoE2 and apoE4 experiments, which

*Figure 5 continued on next page*

*Figure 5 continued*

were extracted, pooled and graphed in *Figure 4e–f* to represent total transport data. Graphs represent mean ±SEM for at least four independent tissues. *, § and # p=0.05, ## and **p=0.01 and ***p=0.001.

DOI: https://doi.org/10.7554/eLife.29595.009

## Circulating HDL in combination with GW3965 induced astrocyte apoE promotes aβ42 transport across tripartite bioengineered cerebral vessels

As the combination of recombinant apoE injected into the anteluminal tissue chamber and HDL injected into the circulating medium significantly increased Aβ40 and Aβ42 transport and reduced Aβ42 tissue accumulation in bipartite vessels (*Figures 4* and *5*), we tested for functional synergy of these lipoproteins in tripartite cerebral vessels by treating the vessels with GW3965 for 72 hr to stimulate native apoE secretion from astrocytes and perfusing HDL through the lumen immediately after Aβ injection. Consistent with our observations in bipartite vessels, neither GW3965 alone nor HDL alone modified the rate of Aβ40 or Aβ42 transport through tripartite vessels over 4 hr (*Figure 7a,b*). However, the combination of GW3965 and HDL together resulted in a significantly increased initial rate of Aβ42 transport without affecting Aβ40 (*Figure 7a,b*). Interestingly, relative to baseline tripartite conditions, neither GW3965, HDL, nor both GW3965 and HDL affected accumulation of Aβ40 or Aβ42 within tripartite tissues after 24 hr (*Figure 7c,d*), which differs from our observations in bipartite vessels (*Figure 5*). To further understand the discrepancy between bipartite and tripartite vessels for Aβ42 tissue accumulation, we hypothesised that basal levels of native apoE secreted from astrocytes in tripartite vessels might already be sufficient to reduce Aβ accumulation. Western blotting confirmed that tripartite tissue lysates had significantly more apoE than bipartite tissue under baseline conditions (*Figure 7e*). We then directly compared Aβ transport and tissue accumulation between bipartite and tripartite vessels and observed increased Aβ42 but not Aβ40 recovery into circulation medium in tripartite compared to bipartite vessels (*Figure 7f,g*). With respect to tissue accumulation, Aβ42 levels were significantly lower and Aβ40 levels showed a trend toward lower levels in tripartite compared to bipartite vessels (*Figure 7h,i*). Notably, Aβ42 levels in tripartite vessels were similar to those observed in bipartite vessels to which recombinant apoE2 was added.

## Pre-aggregation of aβ reduces transport and increases accumulation in bioengineered tripartite vessels similarly to bipartite tissues

We next hypothesised that pre-aggregated amyloid fibrils might accumulate to form CAA in tripartite vessels. To test this, we compared Aβ transport and accumulation by injecting either monomeric or pre-aggregated Aβ40 or Aβ42 on the anteluminal side of tripartite vessels, and observed significantly slower transport of Aβ40 and Aβ42 fibrils compared to monomers (*Figure 8a–b*). Furthermore, tissue accumulation was significantly increased after injection of fibrils compared to monomers (*Figure 8a–b*). To test whether native astrocyte-secreted apoE might reduce aggregation of pre-formed Aβ40 and Aβ42 fibrils (*Castellano et al., 2011*), we quantified Aβ levels in bipartite and tripartite tissues 24 hr after injection of pre-formed Aβ40 and Aβ42 fibrils and observed no significant difference in Aβ accumulation (*Figure 8c–d*). Together, these data suggest that astrocyte-derived apoE specifically facilitates transport of soluble Aβ from the brain.

## Discussion

The profound socioeconomic impact of AD has stimulated extensive research in the last decade, with increased attention on the contribution of cerebrovascular dysfunction in dementia and cognitive decline (*Snyder, 2015*; *Raz et al., 2016*). It is clearly critical to elucidate the mechanisms that regulate Aβ egress from the human brain, as well as understand how to protect cerebrovascular health during aging, yet there are significant barriers toward mechanistic experimentation in a human context. Almost all of our current knowledge about Aβ egress through cerebral vessels has been obtained from animal models, primarily in mice genetically engineered to express human APP, which enables the progressive accumulation of Aβ and β-amyloid to be studied. Despite the tremendous wealth of knowledge generated from animal models, there are considerable challenges in

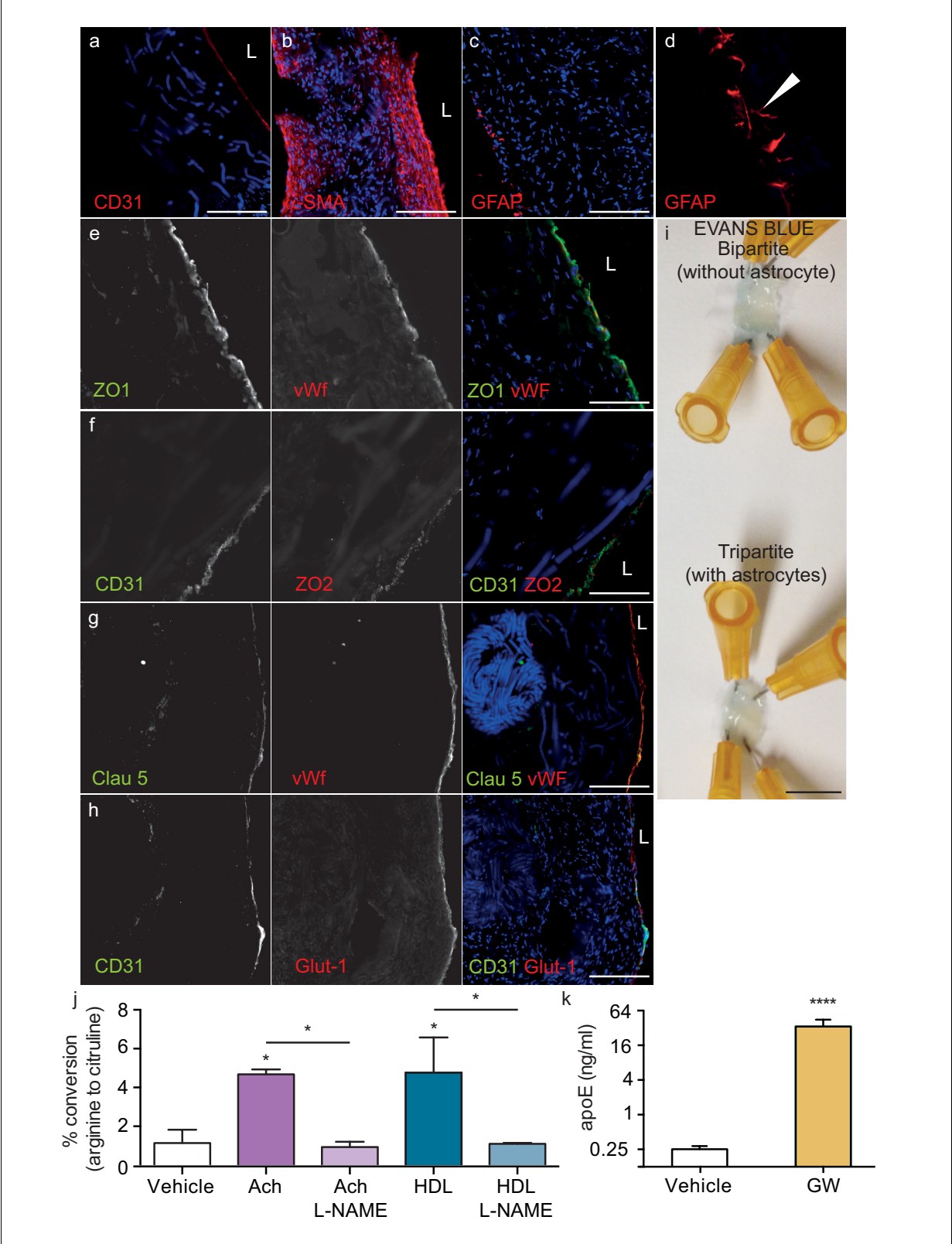

**Figure 6.** Histological structure of tripartite bioengineered vessels. Immunostaining against CD31 confirmed the presence of an EC monolayer on the luminal side of tripartite vessels (**a**), the expression of α-SMA confirmed the smooth muscle phenotype of cells in the inner layers (**b**), and the expression of GFAP confirmed the presence of astrocytes on the anteluminal layers (**c**) of bioengineered tripartite vessels. (**d**) Higher magnification image of a specific GFAP-positive area showing astrocyte end-feet like structures. The EC barrier was further analysed using immunostaining against the tight

*Figure 6 continued on next page*

*Figure 6 continued*

junction proteins ZO-1 (**e**), ZO-2 (**f**) and Claudin 5 (**g**), as well as the BBB-specific glucose transporter (Glut)−1 (**h**). Evans blue staining confirmed a tight endothelium (**i**). EC function was confirmed by measuring NO secretion after either acetylcholine (Ach) or HDL stimulation, in the absence or presence of 1 mM of the eNOS inhibitor L-NAME (**j**). Astrocyte function was confirmed by treating tissues with 1 µM LXR agonist GW3965 for 72 hr and measuring the levels of astrocyte-derived apoE secreted into the medium (**k**). Bars represent 200 µm (**a–h**) or 1 mm (**i**), L: lumen and graphs represent mean ±SEM for at least four independent tissues. *p=0.05, **p=0.01 and ***p=0.001.

DOI: https://doi.org/10.7554/eLife.29595.010

The following figure supplements are available for figure 6:

**Figure supplement 1.** Astrocytes in bioengineered tissues express aquaporin four and NDRG2.

DOI: https://doi.org/10.7554/eLife.29595.011

**Figure supplement 2.** HUVEC express the brain Glut-1 transporter when culture in bioengineered vessels.

DOI: https://doi.org/10.7554/eLife.29595.012

**Figure supplement 3.** Endothelia of bioengineered vessels are impermeable to FITC-dextran.

DOI: https://doi.org/10.7554/eLife.29595.013

translating these results into humans. For example, many AD risk genes have roles in various aspects of lipid metabolism, the most important of these being *APOE* (*Giri et al., 2016*), yet the innate physiological differences between murine and human lipoprotein metabolism (*Getz and Reardon, 2012*) may limit the predictive power of mouse model studies. Although it is well known that humans have three *APOE* allelic variants compared to the single *Apoe* allele in mice, and targeted replacement mice are available that express human *APOE*, there are also other important metabolic distinctions between mice and humans. For example, the primary circulating lipoprotein in rodents is HDL, which, due to its multiple vasoprotective functions, bestows upon mice a natural resilience to cardiovascular diseases such as atherosclerosis. By contrast, the major circulating lipoprotein in humans is LDL, which is mechanistically linked to vascular dysfunction and cardiovascular disease (*Barter et al., 2007*). Despite the wide recognition of the importance of cardiovascular risk factors to AD pathogenesis, they cannot easily or routinely be incorporated into rodent AD models.

In vitro studies using human cells therefore represent a useful alternative approach, yet most studies of cerebrovascular function use monotypic cultures of brain ECs, which do not mimic the complexity of cell-cell and/or cell-matrix interactions found in the native vessel. As an improvement, EC and astrocytes, EC and SMC, and EC and pericytes have been co-cultured, however, this is almost always under static culture conditions (*Di et al., 2009*; *Cho et al., 2007*; *Bicker et al., 2014*; *Navone et al., 2013*; *Man et al., 2008*; *Hatherell et al., 2011*; *Bussolari et al., 1982*; *Cucullo et al., 2007*). More recent studies developed an EC and astrocyte co-culture model using a complex flow system, but this model did not allow histological analysis or cell-ECM interactions to be assessed (*Cucullo et al., 2007*). Considerable recent advances in tissue engineering technology have helped to develop microfluidic systems (i.e. 'organ on a chip') that recapitulate the 3D complexity of the BBB, primarily to produce small vessel structures like capillaries (*Prabhakarpandian et al., 2013*; *Herland et al., 2016*; *Booth and Kim, 2012*; *Griep et al., 2013*; *Cho et al., 2015*). However, as CAA preferentially forms in larger arteries, we aimed to produce 3D in vitro models of functional human vessels that retain the anatomical and functional properties of native human cerebral arteries. In particular, the pulsatile native-like flow environment possible in our 3D models represents a major advantage over studies using static cell culture. Notably, bioengineered peripheral artery equivalents similar to our bipartite model are clinically used to replace damaged vessels in patients with structural cardiovascular diseases (*Schmidt et al., 2006*). Here we demonstrate the utility of bioengineered vessels to investigate how lipoproteins on the brain side or blood side of the vessel affect Aβ transport and development of CAA, using both bipartite and tripartite engineered vessels. Importantly, the generation of a functional, 3D cerebrovascular model using primary human ECs, SMCs and astrocytes represents a major advance in the bioengineering field.

To represent human pial, leptomeningial and penetrating cerebral arteries, we reduced the internal diameter of the scaffold to 2 mm compared to previous engineered vessel equivalents (*Schmidt et al., 2006*; *Robert et al., 2013a*), while retaining the histologically confirmed architecture of native vessels. Specifically, our bipartite vessels contained a monolayer of ECs forming a hollow lumen, surrounded by multiple layers of α-SMA and secreted extracellular matrix components

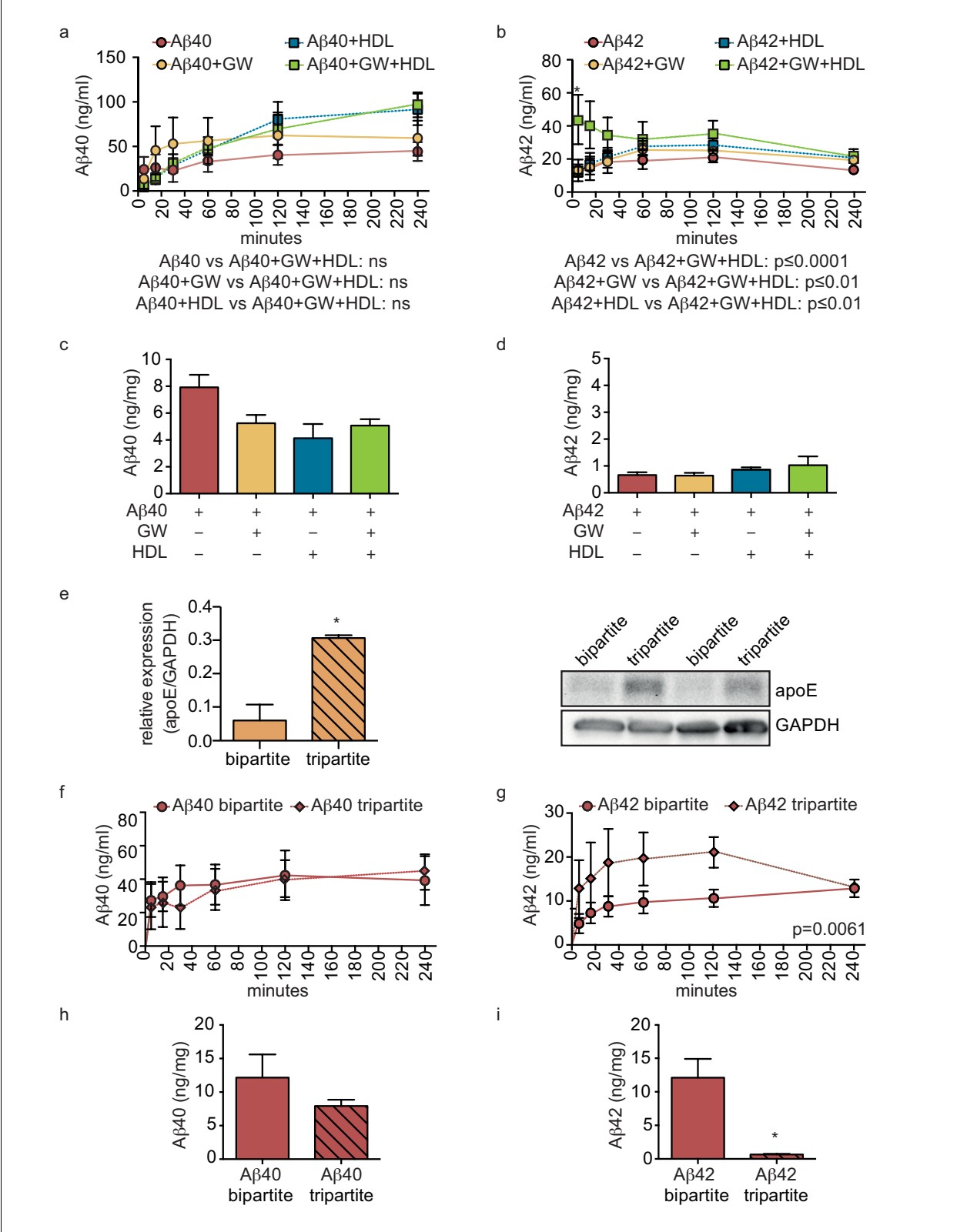

**Figure 7.** HDL facilitates Aβ transport and reduces accumulation in bioengineered tripartite vessels expressing native astrocyte apoE. (a–d) Tripartite vessels were treated with the LXR agonist GW3965 (0.8 μM) for 72 hr to stimulate astrocyte apoE3 secretion. Aβ40 and Aβ42 monomers (1 μM) were injected in the tissue chamber in the absence or presence of 200 μg/ml of circulating HDL, with or without GW3965. The levels of transported Aβ was measured by ELISA from samples collected from the luminal circulating medium at the indicated times over 4 hr (a–b), and from vascular tissue

*Figure 7 continued on next page*

*Figure 7 continued*

collected 24 hr after Aβ injection (**c–d**). ApoE protein level in bioengineered tissues was quantified by Western blotting (**e**) with a representative blot. Aβ transport (**f–g**) and tissue accumulation (**h–i**) were directly compared between bipartite and tripartite bioengineered vessels as above. Graphs represent mean ±SEM for at least four independent tissues. *p=0.05, **p=0.01 and ***p=0.001.

DOI: https://doi.org/10.7554/eLife.29595.014

including collagen and laminin, which demonstrate the in situ functionality of the component cells in the tissue. As a major function of the endothelium is to form a tight barrier between the blood and the interstitial vascular tissue, we also demonstrated the structural integrity of the endothelial layer in bioengineered vessels by demonstrating their impermeability to Evans blue and FITC-Dextran, and confirmed EC function by demonstrating secretion of NO. Furthermore, our completely novel tripartite vessel possessed a layer of GFAP positive astrocytes on the anteluminal side. The formation of structures resembling astrocyte end-feet and secretion of apoE that retains its response to LXR stimulation represent key structural and functional features of astrocytes in healthy cerebral

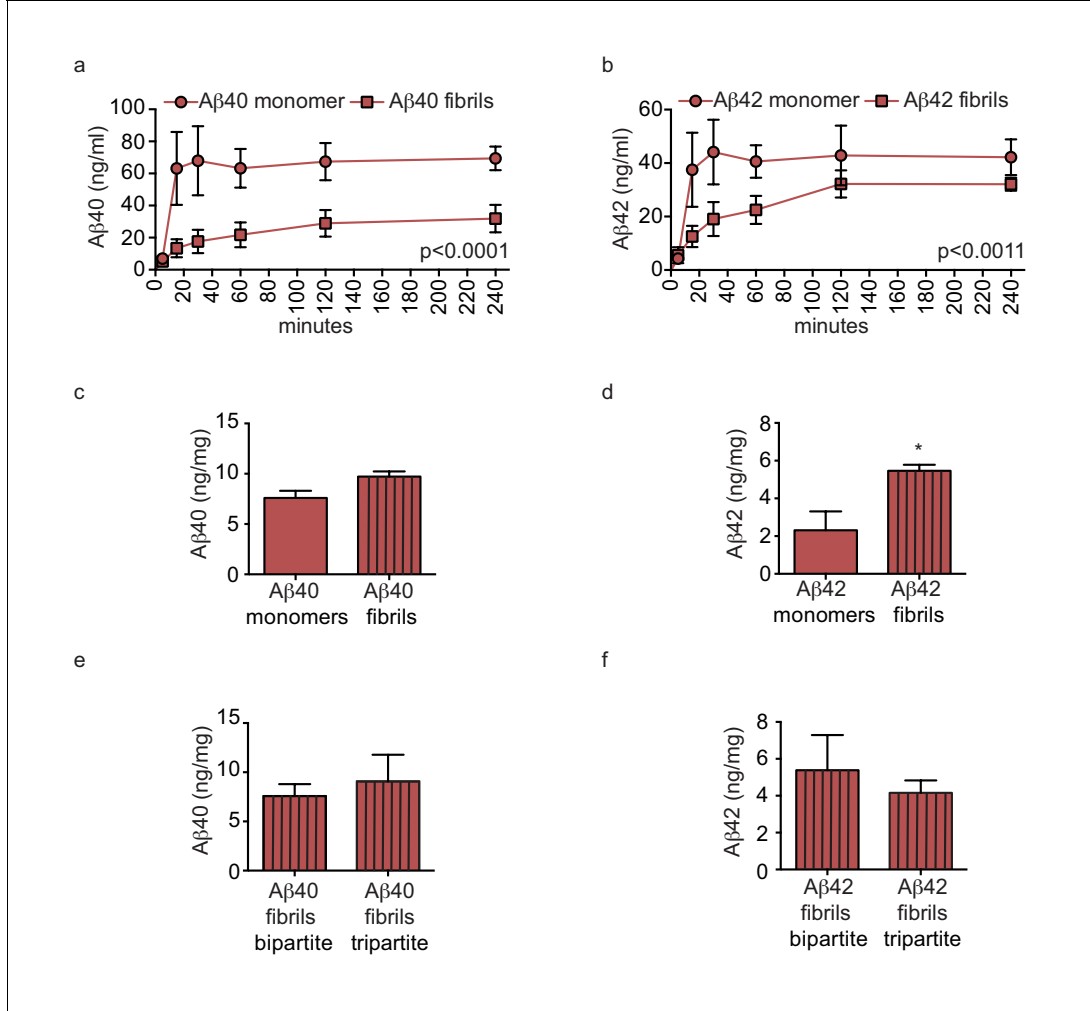

**Figure 8.** Amyloid fibrils are transported more slowly but accumulate similarly between bipartite and tripartite tissues. (**a–d**) Pre-aggregated or monomeric Aβ40 and Aβ42 (1 μM) were injected in the tissue chamber. The levels of transported Aβ were measured by ELISA from samples collected from the luminal circulating medium at the indicated times over 4 hr (**a–b**), and from vascular tissue collected 24 hr after Aβ injection (**c–d**). Aβ tissue accumulation (**e–f**) was directly compared between bipartite and tripartite bioengineered vessels as above. Graphs represent mean ±SEM for at least four independent tissues. *p=0.05, **p=0.01 and ***p=0.001.

DOI: https://doi.org/10.7554/eLife.29595.015

vessels, further supporting the validity of our bioengineered tissue as a valuable new model for mechanistic studies of the human cerebrovasculature. Finally, we observed that the presence of astrocytes induced expression of brain-specific tight junction proteins and transporters in tripartite vessels generated from ECs from a peripheral source, offering the potential to understand how vascular context may reprogram EC phenotype in future studies.

Numerous studies have demonstrated that the accumulation and aggregation of Aβ within the muscular layer of arteries and arterioles represents a key step in the development of CAA (*Biffi and Greenberg, 2011*; *Love, 2009*). We show that our platform can be used to investigate how human lipoproteins on each side of the BBB regulate vascular function and Aβ transport and accumulation. As apoE is the major apolipoprotein expressed in the central nervous system, brain-derived apoE would affect cerebrovascular function from the anteluminal side. By contrast, as apoA-I is produced only in liver and intestine, HDL is found in the circulation and would affect cerebrovascular function from the lumen. Although lipid-free apoA-I can be transported into the brain and is present in CSF (*Stukas et al., 2014a*), there is thus far no evidence that mature HDL might cross the BBB. Our results support a functional cooperation between brain apoE and circulating HDL to promote clearance of Aβ through the cerebral vessel by mechanisms that remain to be fully elucidated. That HDL and apoE consistently affected Aβ42 more than Aβ40 suggests that Aβ40 may be less amenable to lipoprotein-mediated transit across and removal from the vascular wall compared to Aβ42, i.e. Aβ40 is more prone to being retained in the vessel compared to Aβ42. These data are consistent with the observation that Aβ40 is the predominant species found in human CAA (*Yamada, 2015*) even though Aβ42 is reported to be essential for CAA development in mice (*McGowan et al., 2005*). Our results are also consistent with the hypothesized effects of apoE isoform on vascular function, as we demonstrate that recombinant apoE2 promotes more Aβ42 clearance than recombinant apoE4. Importantly, our platform now provides an opportunity to evaluate potential therapeutic strategies to facilitate Aβ clearance, including approaches that target HDL or apoE.

Our study, nevertheless, has several limitations. A major limitation is the availability of primary human astrocytes with distinct *APOE* genotypes, although future studies using standardized astrocytes derived from stem cells could be one possible solution. Similarly, how well recombinant apoE resembles native apoE will require additional investigation especially concerning its lipidation status. Furthermore although previous in vivo studies showed that apoE diminishes Aβ clearance in an isoform-specific manner (*Deane et al., 2008*), these analyses were restricted to the transport of pre-formed Aβ-apoE complexes only and could not differentiate between transport occurring at large vessels versus capillaries. By contrast, our experiments evaluated functional Aβ and apoE interactions under a variety of conditions. Murine and human ECs also differentially clear Aβ with a 30-fold increase of Aβ uptake in mouse cells (*Qosa et al., 2014*). Together, these observations might explain the differences observed between previously published murine data and our human model. A further limitation is that we used readily obtainable umbilical cord cells due to the slower growth rate and supply of primary cerebrovascular cells. Although one could argue that cord-derived cells do not reflect the physiology of the brain vasculature, we clearly demonstrate that HUVEC become reprogrammed to express selective BBB marker proteins when cultured in tripartite bioengineered vessels and that the histological structure of the vessels as well as Aβ accumulation and transport were similar between brain and cord cellular origins. Another limitation is that HDL was obtained from healthy young donors. As HDL functions can be compromised by aging, cardiovascular disease and T2DM (*Riwanto and Landmesser, 2013*), it will be important in the future to understand how HDL purified from aged cognitively healthy individuals, AD subjects, or patients with cardiovascular risk factors may affect cerebrovascular function and Aβ accumulation especially in combination with apoE. It should be noted that the concentration of HDL used in the present study is estimated to be 7-fold lower compared to normolipidomic individuals. HDL concentration within the plasma is typically expressed based on its cholesterol content, with normal levels between 40 and 60 mg/dl, which could roughly be translated to 140 mg protein/dL, similar to apoA-I plasma concentration (*Koren et al., 1985*). We circulated 200 μg/mL HDL in our experiments, similar to what has previously been published for other in vitro studies (*Datta et al., 2001*; *Robert et al., 2013a*). Although previous in vitro studies have also used similar Aβ levels (*Takamatsu et al., 2014*; *Xu et al., 2013*), the Aβ dose used in this study corresponds to a supra physiological concentration of Aβ compared to human brain, CSF or cerebral interstitial fluid, a limitation imposed by the detection limits of our assay (*Brody et al., 2008*; *Seubert et al., 1992*; *Herukka et al., 2015*) As well,

although we could demonstrate that our bioengineered vessels are able to produce NO under physiologically relevant stimuli, our current scaffolding materials are too stiff to permit measurement of vascular compliance. Engineering improvements that explore alternative materials and enable scalable production of bioengineered vessels are also important avenues for future studies. Scalable higher throughput methods are particularly important, as each vessel takes approximately 4 weeks to mature.

In conclusion, we demonstrate for the first time the feasibility to engineer dynamic 3D human artery equivalents to investigate fundamental AD-relevant cerebrovascular processes in vitro, including CAA and the role of lipoproteins in its prevention. Our experimental platform combines the native-like multilayer 3D architecture of both peripheral and cerebral arterial walls with pulsatile flow profiles through a functional lumen, in a manner that permits delivery and sampling of experimental substrates from either side of the bioengineered vessel. By extending the frontier of vascular tissue engineering into diseases involving the cerebrovasculature, our unique cerebral vessel may facilitate in vitro investigations with higher predictive value for human pathologies than current cellular or animal model approaches, as almost all components of the platform can be experimentally manipulated. For example, drug development and delivery studies can be performed, and the effects of hypertension can be studied by modulating factors such as flow rate and pressure of the circulating media. Plasma or immune cells from specific patient groups can be evaluated for effects on cerebrovascular function, including pre- and post-intervention analyses, to better understand the interactions between cardiovascular factors (i.e. T2DM , hypercholesterolemia, hypertension) and brain factors (i.e. *APOE* genotype). Finally, this platform may also facilitate the discovery of blood biomarkers for central nervous system indications. Further advances in bioengineered cerebral vessels to include other brain cells may ultimately improve translational relevance and provide a valuable complement to in vivo studies in animal models.

## Materials and methods

### Cells

All experiments were conducted under an approved clinical protocol (UBC Clinical Ethics Research Board H13-02719) after obtaining written informed consent from all subjects. Human umbilical vein endothelial cells (HUVEC) and human umbilical cord myofibroblasts (UCMFB) were isolated as described (*Robert et al., 2013a*). HUVEC were isolated using the instillation method, where umbilical veins were filled with a solution of collagenase (2 mg/ml, Collagenase A, Roche) in serum-free DMEM (Invitrogen, ThermoFischer Scientific, Waltham, MA). After 20 min at 37°C, Advanced DMEM (Gibco, ThermoFischer Scientific) supplemented with 1% L-glutamine, 0.05% Penicillin/Streptavidin (Pen/Strep) and 10% FBS (Invitrogen) was flushed through the lumen and the cell suspension was centrifuged at 1,200 rpm for 5 min. HUVEC were expanded in endothelial growth medium (EGM−2) (LONZA Inc., Switzerland, supplemented with vascular endothelial growth factor (VEGF), human recombinant insulin-like growth factor-1 (hrIGF-1), human epidermal growth factor (hEGF), amphotericin-B, hydrocortisone, ascorbic acid, heparin, and 2% foetal bovine serum (FBS)) up to passage 7. After HUVEC isolation, the remaining vessels were minced into small pieces (~2–3 mm) and incubated at room temperature (RT) without medium under sterile laminar flow for 25–30 min to ensure physical attachment of UCMFB. Advanced DMEM supplemented with 1% L-glutamine, 0.05% Pen/Strep and 10% FBS was subsequently added to the minced vessels and adherent cells were expanded up to passage 8. Primary mature astrocytes (Sciencell) were cultivated in astrocyte media (Sciencell, Carlsbad, CA) supplemented with astrocyte growth factor, 0.05% Pen/Strep and 2% FBS (Sciencell) up to passage 5. Primary cerebral SMC (Sciencell) were cultivated in Advanced DMEM supplemented with 1% L-glutamine, 0.05% Pen/Strep and 10% FBS up to passage 5. Primary cortical microvascular EC (Cell Systems, Kirkland, WA) were cultivated in complete EGM−2 up to passage 4.

### In vitro fabrication of tissue engineered vascular grafts

Bioengineered constructs were fabricated using a dynamic, semi-pulsatile flow bioreactor system. Tubular biodegradable scaffolds (length 1.5 cm and inner diameter 2 mm) were produced as described (*Robert et al., 2013a*; *Robert et al., 2017*) with minor modifications. Briefly, non-woven polyglycolic acid (PGA, Biomedical Structure, Warwick, RI) meshes (thickness: 1 mm and density: 70

mg/cc) were dip-coated with polycaprolactone (PCL) and polylactate (PLA) by dipping PGA mesh in a solution of 1.75% (w/w) PCL/PLA/tetrahydrofuran (THF) solution (Sigma-Aldrich, St. Louis, MO), shaped into tubes using heat, and externally coated with a 10% PCL/THF (w/w) solution. Scaffolds were sterilized by immersion in 70% ethanol for 30 min followed by three PBS washes and then immersion in advanced DMEM supplemented with 10% FBS for at least 12 hr. UCMFB were seeded at density of $2-3 \times 10^6$ cells/cm$^2$ on the inner surface of the scaffold using fibrin (fibrinogen 10 mg clottable protein/ml PBS and thrombin 100–10 mU/ml PBS) as a cell carrier that was added directly to the scaffold, then incubated under static conditions for a minimum of 3 days before exposure to dynamic flow. The flow of nutrient medium (Advanced DMEM supplemented with 10% FBS, 1% L-glutamine and 0.05% Pen/Strep) was directed through the lumen of the bioreactor circulation loop to mimic blood flow for a minimum of one week. Vascular intermediates were then seeded with HUVEC ($1 \times 10^6$ cells/cm$^2$) and cultivated first in static conditions for a minimum of 5 days in EGM−2 supplemented as above. For bipartite vessels, after the static phase, vascular grafts were placed back in the bioreactor for 14 additional days with increasing medium flow to a final rate of 10 ml/min by the 10th day). For tripartite vessels, after the static phase after HUVEC addition, primary astrocytes were seeded ($1 \times 10^6$ cells/cm$^2$) using fibrin as a cell carrier as above on the antelumen side of the tissue. After 5 min at RT, grafts were placed under flow conditions with EGM-2 supplemented as above in the circulation chamber and complete astrocyte medium in the tissue chamber for 14 additional days with increasing medium flow to 10 ml/min by the 10th day.

## Preparation of beta amyloid peptides

Recombinant Aβ40 and Aβ42 peptides (California Peptide Research, Salt Lake City, UT) were dissolved in hexafluoroisopropanol (HFIP, Sigma-Aldrich). The HFIP was removed by evaporation overnight and stocks were stored at −20°C. On the day of the assay, soluble monomers were prepared by reconstituting the peptide film in DMSO to 5 mM, diluted further to 100 μM in RPMI without FBS. 100 μl of Aβ solution was injected in the tissue chamber containing 900 μl of DMEM (Gibco) without FBS to the desired concentration using a syringe under flow conditions. For fibrils, after reconstitution in RPMI Aβ40 and Aβ42 were incubated at 37°C for 48 hr. Fibrilization was confirmed by dot blot with fibril antibody (OC AB2286 EMD Millipore 1:1000, not shown, RRID: AB_1977024). For luminal recovery, 100 μl circulating medium was collected at the indicated time.

## Lipoproteins

All experiments were conducted under an approved clinical protocol (UBC Clinical Ethics Research Board H14-03357). Upon receipt of written informed consent, 100 ml of fasted blood was collected from normolipidemic healthy donors into vacutainer tubes containing EDTA. Plasma HDL (1.063–1.21 g/ml) was isolated by sequential potassium bromide gradient ultracentrifugation as described (*Robert et al., 2013b*). The purity of the HDL preparations was verified by sodium dodecyl sulfate-polyacrylamide gel electrophoresis (SDS-PAGE) followed by Coomassie blue staining to ensure no LDL or albumin contamination (not shown). Total protein concentration was assessed using the BCA assay (Thermofisher Scientific). Recombinant apoE2 and apoE4 were commercially purchased and solubilized following the manufacture's instructions (ABCAM, apoE2 ab55210, apoE3 ab123764 and apoE4 ab50243). Secretion of endogenous apoE from human astrocytes in tripartite vessels was induced by injecting 0.8 μM GW3965 in the circulation medium 72 hr prior to addition of Aβ. ApoE concentrations were measured using ELISA as previously described (*Fan et al., 2016*). Briefly, ELISA plates were coated overnight with anti-human apoE mAB E276 antibody (MabTech, Cincinnati, OH, RRID: AB_1925746) at 1.55 μg/mL in PBS at 4°C, washed two times with PBST (0.05% Tween 20 in PBS), and blocked with 0.1% Blocker A (MesoScale Discovery, Rockville, MA) in PBST. After 1 hr incubation at RT and two washes with PBST, medium or human recombinant ApoE standard (Mab-Tech) were added to each well. After 1 hr at RT and two subsequent PBST washes, biotinylated anti-human apoE monoclonal antibody E887 (MabTech, RRID: AB_1925729) was added to each well at a concentration of 0.5 μg/mL in blocking buffer. After 1 hr at RT, plates were washed before adding QuantaBlue Substrate (Pierce, ThermoFischer Scientific) working solution (9 parts of Substrate Solution to one part Stable Peroxide Solution). Fluorescence was read after 15 min at RT on an EnSpire 2300 Multilabel Plate Reader ($325_{Ex}/420_{Em}$).

## Endothelium integrity

Evans blue (Sigma-Aldrich) was injected at a final concentration of 0.5% in the circulation loop of the bioreactor for 10 min followed by continuous PBS washing for 20 min. Vessels were cut open longitudinally and en face preparations were analysed macroscopically with photo documentation. Restriction of paracellular transport was determined by measuring FITC dextran extravasation to the tissue chamber as described (*Gaillard et al., 2001*). Briefly 250 µg/ml of 4 kDa or 40 kDa FITC-dextran (Sigma-Aldrich) was circulated through the lumen of bipartite, tripartite tissues or scaffold only. After 1 hr tissue media was collected, fluorescent was read at RT on an EnSpire 2300 Multilabel Plate Reader ($492_{Ex}/518_{Em}$) and the permeability coefficient ($P_{app}$) was calculated using the following equation: $P_{app}=(dQ/dt)*(1/A*C_0*60)$ where $dQ/dt$ is the amount of FITC-dextran transported per minute (ng/min), $A$ is the surface area of the tissue (cm [*Attems and Jellinger, 2014*]), $C_0$ is the initial concentration of FITC-Dextran (ng/ml) and 60 is the conversion from minutes to seconds.

## NO measurement

NO synthesis was measured as described (*Robert et al., 2013a*) using a commercial NOS activity assay kit (Caymenchemical, Ann Arbor, MI). Briefly, a 2–3 mm ring of vascular tissue was mechanically ground in 150 µl of ice cold homogenized buffer (25 mM Tris-Cl, pH7.4, 1 mM EDTA and 1 mM EGTA) and centrifuged 15 min at 4°C at 10,000 g. The supernatant was aliquoted and incubated in reaction buffer containing 25 mM Tris-Cl, pH7.4, 0.25 mM EDTA, 0.6 mM $CaCl_2$, 1 mM NADP, 200 nM calmodulin, 3 µM tetrahydrobiopterin, 1 µM flavin adenine dinucleotide, 1 µM flavin adenine monoucleotide and 0.2 µCi L-$^3$H-arginine; PerkinElmer, Waltham, MA) in the presence of 10 nM acetylcholine (Ach), 0.2 mg/ml HDL or 1 mM L-NG-nitroarginine methyl ester (L-NAME). After 60 min at 37°C, the reaction was stopped by adding 400 µl of stop buffer (50 nM HEPES, 5 mM EDTA, pH 5.5). The solution was loaded onto an ion exchange column equilibrated with stop buffer to separate L-$^3$H-citruline from L-$^3$H-arginine. Scintillation mix (Ultimate Gold, PerkinElmer) was added to the supernatant and counted using LS6500 β-counter (Beckman Coulter, Brea, CA). The percent citrulline formed was calculated as follows: % conversion = (cpm reaction-cpm background)/cpm total *100.

## Histology and immunohistochemistry of bioengineered vessels

For standard histology, bioengineered vessels were fixed in formalin (Thermo Fisher Scientific) for 24 hr, dehydrated through a series of graded ethanol series in a tissue processor (Sakura, Torrance, CA), embedded in paraffin and sectioned at 7 µm thickness. For staining, sections were deparaffinised in 3 baths of Xylene and rehydrated through a graded ethanol series (100, 90, 80% and 70%, for 1 min). Sections were stained using Haematoxylin and Eosin (Sigma-Aldrich) and Picrosirius (ABCAM, Canada) following the manufactures' instructions. For immunohistochemistry and Thioflavin-S staining, vessels were washed twice in PBS, cryopreserved in O.C.T. embedding matrix, and processed on a cryotome to generate 20 µm sections that were stored at –80°C until further analysis. Sections were rehydrated in PBS for 2 × 10 min before fixing in 4% paraformaldehyde (PFA) for 20 min at RT. After one Tris-HCl (0.5 mM pH 7.6) and two PBS washes, sections were blocked for 30 min in 5% goat serum and 1% BSA in PBS. For immunofluorescence, sections were incubated overnight at 4°C with specific antibodies against CD31 (WM59 Biolegend, San Diego, CA, 1:50, RRID: AB_314328), von Willebrand factor (Sigma-Aldrich, 1:200, RRID: AB_259543), α-SM-actin (1A4 Sigma-Aldrich, 1:200, RRID: AB_476856), collagen IV (EMD Millipore, 1:100, RRID: AB_2276457), laminin (Abcam, 1:200, RRID: AB_298179), claudin 5 (4C3C2 ThermoFisher Scientific, 1:50, RRID:AB_2533200), ZO-1 (1A12 ThermoFisher Scientific, 1:50, RRID: AB_2533147), ZO-2 (ThermoFisher Scientific, 1:50, RRID: AB_2533976), Aβ 1–16 (6E10 ThermoFisher Scientific, 1:50, RRID: AB_2565328) and Aβ fibrils (OC AB2286 EMD Millipore 1:200, RRID: AB_1977024). After three additional PBS washes, sections were incubated for 45 min at RT with anti-rabbit or anti-mouse Alexa-488 or Alex-594 secondary antibodies (Invitrogen) with DAPI. Sections were finally washed three times in PBS and mounted in Prolong Diamond antifade (ThermoFisher Scientific). For Thioflavin-S staining, sections were rehydrated and stained in 1% Thioflavin-S (Sigma-Aldrich) for 10 min at RT in the dark, washed five times with PBS and mounted in Vectashield (Vector). Brighfield and fluorescent images were acquired with an inverted microscope (Zeiss, Germany) or a SP8 confocal microscope (Leica, Canada).

## Aβ quantification

Luminal medium was collected from the circulation chamber and 5 mm tissue rings were crushed and lysed in RIPA buffer (10 mM Tris pH 7.4, 150 mM NaCl, 1.0% NP-40, 1.0% sodium deoxycholate, 0.1% SDS and cOmplete protease inhibitor with EDTA (Roche, Switzerland)). Aβ40 (KHB3442, Life Tech, ThermoFischer Scientific) and Aβ42 (KHB3482, Life Tech) were quantified using commercial ELISAs and normalized to total protein concentration, measured by BCA.

## Thioflavin-T quantification

Aβ fibrillization was measured as described (*Truran et al., 2016*). Briefly, 10 µl of RIPA homogenized tissue was mixed with 90 µl of 20 µM Thioflavin-T (Sigma-Aldrich) in 150 mM NaCl with 5 mM HEPES, pH 7.4 in black 96-well plates. Formation of fibrillar β-amyloid pleated sheets was monitored by excitation at 440 nm and measuring emission intensity at 490 nm using an Infinite M2000 Pro plate reader (Tecan).

## Human tissues

Human cortical brain tissues were previously obtained from the Brain and Tissue Bank, University of Maryland School of Medicine under the UBC clinical protocol (C04-0595). Human umbilical cords were obtained from the British Columbia Women's Hospital, Vancouver, BC, Canada.

## Statistical analyses

Statistical comparisons between different groups were performed using Student T-test, one way ANOVA with Dunnett post test, or two way ANOVA with Sidak multi comparison test. Data were obtained from at least four independently generated bioengineered vessels and graphically represented as mean ±standard error of the mean (SEM). P-values of <0.05 were considered statistically significant. All statistical analyses were performed using GraphPad Prism-5 software (RRID:SCR_002798).

## Acknowledgements

This work was supported by operating grants from Canadian Institutes of Health Research (CIHR), the Canadian Consortium of Neurodegeneration and Aging (CCNA), a Djavad Mowafaghian Centre for Brain Health Catalyst grant, philanthropic funding from the Jack Brown and Family Alzheimer's Research Foundation, the YP Heung Foundation, a private BC-based foundation to CLW, and a Weston Brain Institute Rapid Response grant to JR JR was further supported by a BrightFocus post-doctoral fellowship and a Swiss National Science Foundation Early postdoctoral fellowship. EBB was supported by a CIHR Doctoral Scholarship. IK was supported by a Weston Brain Institute Rapid Response grant.

## Additional information

### Funding

| Funder | Author |
| --- | --- |
| Weston Brain Institute Rapid Response | Jerome Robert |
| BrightFocus Foundation | Jerome Robert |
| Swiss National Science Foundation | Jerome Robert |
| Canadian Institutes of Health Research | Emily B Button |
| University of British Columbia | Emily B Button |
| Canadian Institutes of Health Research | Cheryl L Wellington |
| Canadian Consortium of Neurodegeneration and Aging | Cheryl L Wellington |

| Djavad Mowafaghian Center for Brain Health Catalyst Grant | Cheryl L Wellington |
| Jack Brown and Family Alzheimer's Research Foundation | Cheryl L Wellington |
| Y.P. Heung Foundation | Cheryl L Wellington |
| Weston Brain Institute Rapid Response | Iva Kulic |

The funders had no role in study design, data collection and interpretation, or the decision to submit the work for publication.

## Author contributions

Jerome Robert, Conceptualization, Data curation, Formal analysis, Funding acquisition, Investigation, Methodology, Writing—original draft, Writing—review and editing; Emily B Button, Brian Yuen, Megan Gilmour, Kevin Kang, Arvin Bahrabadi, Wenchen Zhao, Data curation; Sophie Stukas, Iva Kulic, Data curation, Investigation; Cheryl L Wellington, Conceptualization, Resources, Supervision, Funding acquisition, Investigation, Project administration, Writing—review and editing

## Author ORCIDs

Jerome Robert http://orcid.org/0000-0002-2847-9362
Emily B Button http://orcid.org/0000-0003-0390-0867
Kevin Kang http://orcid.org/0000-0002-5929-1645
Arvin Bahrabadi http://orcid.org/0000-0003-1042-6075
Iva Kulic http://orcid.org/0000-0001-9093-113X
Cheryl L Wellington http://orcid.org/0000-0001-7014-039X

## Ethics

Human subjects: All experiments with umbilical cells were conducted under an approved clinical protocol (UBC Clinical Ethics Research Board H13-02719) after obtaining written informed consent from all subjects. All experiments involving blood were conducted under an approved clinical protocol (UBC Clinical Ethics Research Board H14-03357) with written consent from the donors.

## Decision letter and Author response

Decision letter https://doi.org/10.7554/eLife.29595.019
Author response https://doi.org/10.7554/eLife.29595.018

# Additional files

## Supplementary files

• Transparent reporting form
DOI: https://doi.org/10.7554/eLife.29595.016

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
