## [Decision Letter]

Thank you for submitting your article "Clearance of β-amyloid is facilitated by apoE and circulating high-density lipoprotein in bioengineered human vessels" for consideration by *eLife*. Your article has been reviewed by two peer reviewers, and the evaluation has been overseen by a Reviewing Editor and a Senior Editor. The reviewers have opted to remain anonymous.

The reviewers have discussed the reviews with one another and the Reviewing Editor has drafted this decision to help you prepare a revised submission.

Summary:

In this manuscript Robert et al. describe the development of a three dimensional model of a blood vessel resembling that seen in the brain using bioengineered human blood vessels. They use this model to study transport kinetics of Abeta from the "tissue side" across blood vessels into the luminal side and the effect of apoE isoform as well as circulating HDL on the clearance rate of Abeta40 and Abeta42. They perform this analysis with both monomeric and fibrillar abeta.

The authors report that ApoE2 is more effective at aiding transport of both Abeta isoforms across blood vessels and that HDL and ApoE work synergistically to make this transport even more effective. Overall this is a robust new in vitro model of blood vessels resembling that seen in brain that can be used to study a wide variety of questions related to CAA, AD and the complex interplay between cardiovascular health and neurodegeneration. While there are some differences with native cerebral arterioles such as using human umbilical derived cells, this is the most physiological in vitro model produced to date that has been developed to study processes related to CAA, AD, and Abeta accumulation and transport to blood vessels.

Both reviewers and the Reviewing Editor believe that this is an excellent pioneering study that has the potential to revolutionize the mechanistic exploration of amyloid export from the brain across the vasculature and for drug discovery aimed at improving this process. The novelty, the adaptability of the tissue engineering to various genetic constellations and the quality of the data in addition to the high translational value of the method impose an extraordinarily strong suitability for publication in *eLife*.

Essential revisions:

1) Figure 1 is missing.

2) Regarding transport experiments performed in Figure 3 and beyond, the physiological level of the two Abeta species studied in a normal human/mouse brain is much lower than what is utilized in the manuscript. One μM is supra physiological. It is likely levels like this are required to be able to detect the Abeta. It would be helpful if the authors could point this out as a caveat/shortcoming in the Discussion.

---

## [Author Response]

Essential revisions:1) Figure 1 is missing.

We apologize for the oversight and thank the reviewers for noticing the missing Figure 1. We loaded the missing figure, which depicts the schematic representation of the bipartite and tripartite bioengineered vessels.

2) Regarding transport experiments performed in Figure 3 and beyond, the physiological level of the two Abeta species studied in a normal human/mouse brain is much lower than what is utilized in the manuscript. One μM is supra physiological. It is likely levels like this are required to be able to detect the Abeta. It would be helpful if the authors could point this out as a caveat/shortcoming in the Discussion.

We agree with this point. Although local Aβ concentration is difficult to estimate, the concentration used in the present manuscript is indeed supra physiological compared to normal human brain, CSF and cerebral interstitial fluid ^1–3^. We were limited to using 1 μM throughout the manuscript to allow quantification of Aβ especially in tripartite vessels or in the presence of lipoproteins where Ab levels were approaching the limit of detection of the ELISA assay used in our study.

It is also important to note that the HDL concentration used in the manuscript is below normolipidomic plasma concentration. HDL concentration within the plasma is typically expressed as HDL cholesterol content, with normal levels between 40 and 60 mg/dl, which could roughly be translated to 140 mg protein/dL, similar to the apoA-I plasma concentration ^4^. The concentration of HDL used in our study (200 ug/mL) is therefore 7-fold lower than the typical plasma concentration, but is consistent to many previously published in vitro studies due to the limitation of the availability of HDL. These two points have been emphasized in the Discussion as potential caveats of our study as follows:

“Another limitation is that HDL was obtained from healthy young donors. As HDL functions can be compromised by aging, cardiovascular disease and type II diabetes mellitus ^5^, it will be important in the future to understand how HDL purified from aged cognitively healthy individuals, AD subjects, or patients with cardiovascular risk factors may affect cerebrovascular function and Aβ accumulation especially in combination with apoE. […]As well, although we could demonstrate that our bioengineered vessels are able to produce NO under physiologically…”

References

1) Brody, D. L. et al. Amyloid-beta dynamics correlate with neurological status in the injured human brain. Science (New York, N.Y.) 321, 1221–4 (2008).

2) Seubert, P. et al. Isolation and quantification of soluble Alzheimer’s β-peptide from biological fluids. Nature 359, 325–327 (1992).

3) Herukka, S.-K. et al. Amyloid-β and Tau Dynamics in Human Brain Interstitial Fluid in Patients with Suspected Normal Pressure Hydrocephalus. Journal of Alzheimer’s disease : JAD 46, 261–9 (2015).

4) Koren, E., Puchois, P., McConathy, W. J., Fesmire, J. D. & Alaupovic, P. Quantitative determination of human plasma apolipoprotein A-I by a noncompetitive enzyme-linked immunosorbent assay. Clinica chimica acta; international journal of clinical chemistry 147, 85–95 (1985).

5) Riwanto, M. & Landmesser, U. High-density lipoproteins and Endothelial Functions: Mechanistic Insights and Alterations in Cardiovascular Disease. Journal of lipid research (2013). doi:10.1194/jlr.R037762